# Determinants of COVID-19 Vaccine Uptake and Hesitancy among Healthcare Workers in Tanzania: A Mixed-Methods Study

Maryam A. Amour [1], Innocent B. Mboya [2,3,*], Harrieth P. Ndumwa [1], James T. Kengia [4], Emmy Metta [5], Belinda J. Njiro [1], Kasusu Klint Nyamuryekung'e [6], Lwidiko E. Mhamilawa [7,8], Elizabeth H. Shayo [9], Frida Ngalesoni [10], Ntuli Kapologwe [4], Bruno Sunguya [1], Sia E. Msuya [3,11] and Albino Kalolo [12]

1 Department of Community Health, School of Public Health and Social Sciences, Muhimbili University of Health and Allied Sciences, Dar es Salaam P.O. Box 65001, Tanzania; maryam.a.amur@gmail.com (M.A.A.)
2 Department of Translational Medicine, Lund University, 202 13 Malmö, Sweden
3 Department of Epidemiology and Biostatistics, Institute of Public Health, Kilimanjaro Christian Medical University College, Moshi P.O. Box 2240, Tanzania
4 President's Office Regional Administration and Local Government, Dodoma P.O. Box 1923, Tanzania
5 Department of Behavioural Sciences, School of Public Health and Social Sciences, Muhimbili University of Health and Allied Sciences, Dar es Salaam P.O. Box 65001, Tanzania
6 Department of Community Dentistry, School of Dentistry, Muhimbili University of Health and Allied Sciences, Dar es Salaam P.O. Box 65014, Tanzania
7 Department of Parasitology and Medical Entomology, Muhimbili University of Health and Allied Sciences, Dar es Salaam P.O. Box 65001, Tanzania
8 Department of Women's and Children's Health, International Maternal and Child Health, Uppsala University, 751 85 Uppsala, Sweden
9 National Institute for Medical Research, Dar es Salaam P.O. Box 9653, Tanzania
10 Amref Health Africa in Tanzania, Dar es Salaam P.O. Box 2773, Tanzania
11 Department of Community Medicine, Kilimanjaro Christian Medical Centre, Moshi P.O. Box 3010, Tanzania
12 Department of Public Health, St. Francis University College of Health, and Allied Sciences, Morogoro P.O. Box 175, Tanzania
* Correspondence: innocent.mboya@med.lu.se

**Abstract:** The novel Coronavirus disease 2019 (COVID-19) presents a major threat to public health but can be prevented by safe and effective COVID-19 vaccines. Vaccine acceptance among healthcare workers (HCWs) is essential to promote uptake. This study, aimed to determine the COVID-19 vaccination uptake and hesitancy and its associated factors among HCWs in Tanzania. We employed a convergent-parallel mixed-methods design among 1368 HCWs across health facilities in seven geographical zones in Tanzania in 2021. We collected quantitative data by using an interviewer-administered questionnaire and qualitative data, using in-depth interviews and focus group discussions. Participants in the quantitative aspect were conveniently selected whereas those in the qualitative aspect were purposively selected based on their role in patient care, management, and vaccine provision. Stata software version 16.1 was used in the analysis of quantitative data and thematic analysis for the qualitative data. Multiple logistic regression was used to assess the determinants of COVID-19 vaccine uptake. The median age of 1368 HCWs was 33, and the interquartile range was 28–43 years; 65.6% were aged 30+ years, and 60.1% were females. Over half (53.4%) of all HCWs received the COVID-19 vaccine, 33.6% completely refused, and 13% chose to wait. HCWs aged 40+ years, from lower-level facilities (district hospitals and health centers), who worked 6+ years, and with perceived high/very high risk of COVID-19 infection had significantly higher odds of vaccine uptake. The qualitative data revealed misinformation and inadequate knowledge about COVID-19 vaccine safety and efficacy as the key barriers to uptake. Nearly half of all HCWs in Tanzania are still unvaccinated against COVID-19. The predominance of contextual influence on COVID-19 vaccine uptake calls for interventions to focus on addressing contextual determinants, focusing on younger HCWs' population, short working duration, those working at different facility levels, and providing adequate vaccine knowledge.

**Keywords:** COVID-19; vaccine hesitancy; vaccine acceptability; vaccine uptake; barriers; healthcare workers; Tanzania

## 1. Introduction

The novel Coronavirus disease 2019 (COVID-19) presents a major threat to public health and was declared a global pandemic by the World Health Organization (WHO) on 11 March 2020 [1]. Since the first COVID-19 case in China, there have been increased efforts to delineate its spread, and different strategies to prevent transmission and treat the disease have been employed worldwide [2].

Until 12 April 2023, the WHO reported 762,791,152 confirmed COVID-19 cases globally, with 6,897,025 deaths [3]. In the context of Tanzania, the government registered its first case of COVID-19 on 16 March 2020, and from 3 January 2020 to 26 August 2022, Tanzania reported 38,712 confirmed cases of COVID-19, with 841 deaths [4]. Effective COVID-19 vaccination is a critical and top-priority step to stop the pandemic and slow the spread of SARS-CoV-2 infection [5]. The WHO reported that, until May 2022, only 57 countries had vaccinated 70% of their population, and almost all of them were high-income countries [3]. The first consignment with 1,058,450 doses of the Janssen COVID-19 vaccines arrived in Tanzania on 24 July 2021, through the COVAX arrangement targeting frontline healthcare workers (HCWs), older people, and individuals with comorbid conditions. By August 2022, a total of 22,082,377 vaccine doses were administered in the country, accounting for about 37% of the population [4].

Like other countries, Tanzania had HCWs as one of the priority groups to receive vaccination owing to their regular contact with patients, including COVID-19 patients. The vaccination of HCWs for other infectious diseases, e.g., Hepatitis B, in general, has also been advocated by the WHO for the same reasons. In the case of COVID-19, HCWs play a key role in recommending the vaccine to their patients, helping in the vaccination process, and role-modeling preventive measures against COVID-19 transmission, hence reducing its burden globally [6].

Globally, the uptake of vaccines in an emergency setting, vaccine efficacy, and potential side effects have been a concern [6]. COVID-19 vaccine uptake among HCWs has ranged between 52% in Malta [7] to as high as 92% in Germany [8,9]. In Africa, recent studies have reported poor attitudes toward vaccine acceptance among HCWs, with only 42.3% having a positive attitude toward the same [10]. Myths and misconceptions about the vaccine have been observed in many African countries as well [11]. Several challenges exist concerning COVID-19 data and vaccine acceptance in Tanzania [12]. For example, initially, the Government of Tanzania took a different approach to tackling COVID-19 which did not prioritize vaccines as one of the preventive measures [13]. Several months after the first COVID case, a national COVID-19 committee was formed, and they recommended the government engage in contingency and response plans for COVID-19 at all levels to provide reports on the COVID-19 pandemic status in the country and strengthen protective measures to prevent another wave of the disease. There are no studies that were conducted in Tanzania to determine vaccine acceptance and associated factors among HCWs in Tanzania. The perceptions and acceptance of these vaccines in Tanzania must be understood so that appropriate interventions are put in place toward promoting COVID-19 vaccine uptake in the country. This study, therefore, aimed to determine the COVID-19 vaccination uptake and hesitancy and its associated factors among HCWs in Tanzania.

## 2. Methods

### 2.1. Study Design

This study employed a convergent-parallel mixed-methods design [14]. The convergent-parallel component refers to the collection and analysis of the two independent strands of quantitative and qualitative data in a single phase. In this study, quantitative and qualita-

tive were taken to hold equal weight, and the results from the two strands of analysis were merged to look for convergence, divergence, contradictions, and relationships [14,15]. A mixed-methods approach was applied in this study because one single method would not be sufficient to capture the holistic picture of the determinants of COVID-19 vaccine uptake. This approach was set to allow us to explain the results in more detail, integrating multiple perspectives and identifying different plausible causal pathways between the determinants and the COVID-19 vaccine uptake. The Good Reporting of A Mixed Methods Study (GRAMMS) framework for reporting mixed-methods studies was followed in reporting the results of this study [16]; i.e., quantitative and qualitative data were collected concurrently among HCWs in Mainland Tanzania. The quantitative aspect aimed at determining HCWs' uptake of COVID-19 vaccines. The qualitative approach further explored the motives or barriers for vaccine uptake and thus provided a deeper understanding of the subject.

*2.2. Study Setting*

The quantitative survey was conducted among HCWs working in health facilities located within the seven geographical zones (Central, Coastal, Lake, Northern, Southern Highlands, and Western zone) in Tanzania [17]. One region was selected randomly among all regions in the seven geographical zones. The seven geographical zones were drawn from the United Republic of Tanzania, with an estimated population of 61,627,284 people. The GDP grew at 4.9% in 2021, up from 4.8% in 2020, but there were notable economic downsides of COVID-19 that affected the national workforce's productivity [18].

The qualitative survey was conducted in four (4) regions of Tanzania. Two regions were selected based on COVID-19 prevalence: one with high and one with comparatively low prevalence. The remaining two regions were selected from the poorly performing regions in COVID-19 vaccination uptake.

*2.3. Study Population, Sample Sizes, and Sampling Techniques*

*Quantitative component*: The study population for the quantitative component comprised all working HCWs from district hospitals, health centers, and Regional Referral Hospitals (RRHs) across all cadres of healthcare service provision in Tanzania, including medical, dental, laboratory, pharmaceutical, nursing, physiotherapists, and orthotics. It also included social workers and community health workers situated in selected health facilities at the national, zonal, and regional referral hospitals, district hospitals, and health centers. The sample size calculation was performed using a single proportion formula, taking a standard normal value of 1.96 under the 95% confidence limit, 50% proportion of vaccine hesitancy (for maximization of sample size), and 1.5 design effect to address the clustering effect. Using a 3.5% margin of error, we estimated a minimum sample size of 1167. Adjusting for a non-response rate of 20%, the minimum sample of 1400 HCWs was to participate in this study.

The survey was conducted from October 2021 to November 2021. HCWs were randomly selected from health facilities providing COVID-19 vaccines in the country. Seven (7) regional referral hospitals, fourteen (14) district hospitals, and twenty-one (21) health centers were included in the quantitative aspect. From the selected health facilities, convenient sampling was used to select eligible HCWs within facilities considering a proportionate representation of all cadres available in selected health facilities.

*Qualitative component*: The study population included regional and district outbreak response teams, regional and district cold chain coordinators, and health facility in-charges purposively recruited based on their role in patient care and management, as well as vaccine provision. Moreover, the study included a purposeful sample of HCWs from health facilities within each of the four regions (comprehensive details of the level of sampling and types of involved HCWs are provided as an additional file). The qualitative component included twenty-four in-depth interviews (IDIs), five focus group discussions (FGDs), and seven group interviews in four regions. IDI, group interviews, and FGDs participants were

purposively recruited based on the virtue of their knowledge, experiences, or position regarding COVID-19 vaccine uptake and its related context [19].

### 2.4. Data Collection Tools and Procedures

*Quantitative data collection*: Quantitative data were collected using a validated interviewer-guided questionnaire through the KoboToolbox [20]. The questionnaire had questions developed based on various studies, and the WHO proposed questions to assess vaccine uptake and hesitancy [21–23]. The tools were developed in English and translated into the Swahili language, the lingua Franca of Tanzania. For validation purposes and to ensure that all questions were clearly understood and consistently answered by all respondents across regions given the geographical and cultural variability, a pilot testing of the tools was conducted in selected facilities other than those included in this study. All data were collected by trained research assistants (RAs). During data collection, the RAs visited the respective healthcare facilities, introduced themselves, and explained the study purpose to the HCWs. Consent information was then administered in a quiet place around the health facility, with special emphasis placed on issues of anonymity and confidentiality to encourage truthful responses. Only the consenting individuals were interviewed.

*Qualitative data collection*: Qualitative data collection was performed using semi-structured interview guides for IDI, FGD, and group interviews. IDIs were conducted for key officials at regional and district levels and those in charge of health facilities. IDIs at the regional and district level included members from the Regional/Council Health Management Team (R/CHMT) like the Regional Medical Officer (RMO), Regional Vaccination Officer (RVO), District Medical Officer (DMO), District Vaccination Officer (DVO), and from the epidemic response and vaccine committees and in-charges from the health facilities administering the COVID-19 vaccines. These IDIs explored perceptions of HCWs toward COVID-19 vaccines, motives among vaccinated individuals, and barriers to vaccination uptake.

Group interviews and FGDs were conducted with HCWs in the selected districts within the study regions. FGDs were composed of HCWs of different cadres and comprised between 6 and 12 participants. FGDs were conducted among HCWs working at 8 selected health facilities in each of the four regions. The discussions explored opinions on COVID-19 vaccine acceptance and the discussants' willingness to administer the vaccines to themselves and to advocate its uptake to other stakeholders. HCWs were also requested to advise possible ways to improve vaccine uptake. Trained research assistants collected data. All interviews were conducted in Swahili and audio recorded with the permission of the study participants. Field notes, as a reflective diary, were maintained for the enhancement of credibility.

### 2.5. Study Variables

The primary outcome variable was the uptake of COVID-19 vaccines. The secondary outcome was hesitancy toward COVID-19 vaccination. This was measured by asking the participants whether they have taken the COVID-19 vaccine or if they will delay taking the vaccine; the responses were "have already taken the vaccine", "will wait for some time", or "will not take the vaccine at all". The last two options constituted vaccine hesitancy [24]. The explanatory variables included participant sociodemographic characteristics, awareness, and knowledge about COVID-19 vaccines, perception of COVID-19, social norms and networks, vaccine misinformation, and vaccine accessibility. Sociodemographic characteristics included the region of residence, health facility level and ownership (public or private), sex (male and female), age in years, education level, duration working in years, and cadre (physicians, nurse/midwife, pharmacy, laboratory, admin/supporting staff, and others). Regarding education level, primary/secondary school education is the first stage of formal education in Tanzania, corresponding to elementary school, middle school, and high school. Certificate education involves short-term courses that provide specialized training and skills in a particular area of study. Diploma education is more comprehensive

than certificate programs and is offered by vocational schools, community colleges, or technical institutes to provide practical training in a specific field. In this study, a physician is any individual with a medical degree, diploma, or equivalent qualification.

### 2.6. Data Analysis

*Quantitative data analysis*: The quantitative data were transferred from the Kobo Toolbox to an Excel spreadsheet. The data was then transferred to Stata software version 16.1 (College Station, TX, USA) for cleaning and analysis. Descriptive statistics summarized the data, specifically frequencies, and percentages for categorical data and means/medians with standard deviations/interquartile range for numerical data. The chi-squared test compared the proportions of COVID-19 vaccine acceptance by HCWs' characteristics. Logistic regression models were used to determine factors associated with COVID-19 vaccine acceptance among HCWs in Tanzania. The selection of variables for inclusion in the adjusted/multivariable analysis was informed by the previous literature. In addition, a *p*-value of <0.05 was considered statistically significant in the descriptive and inferential analysis.

*Qualitative data analysis*: The audio-recorded IDIs and FGDs were transcribed verbatim into Word file documents, and non-verbal cues were accounted for. The transcription process started within 24 h after the interview to allow for follow-up on issues for more clarity and the determination of data saturation. The transcribed transcripts were checked against the audio records by two of the research team members to ensure the accuracy and quality of the generated data. Based on grounded theory, the team used thematic analysis to analyze the information, following the five stages as described by Braun and Clarke (2014) [25] to establish meaningful patterns in the data: familiarization with the data, generating initial codes, searching for themes among codes, reviewing themes, and presenting the results. Similarities and differences were observed during the analysis. In illustrating some of the themes to answer the key research questions, participants' own words (quotes) are used.

### 2.7. Integrated Interpretation

To obtain a holistic and cross-validated picture of the determinants of COVID-19 vaccine uptake, we integrated quantitative and qualitative findings. The integration occurred at the interpretation level, whereby a triangulation approach was used. Since quantitative and qualitative findings held equal weight and were used to explore different aspects of the same phenomena, our integrated interpretation focused on looking for where the findings converged, offered complementary information or appeared to contradict one another. This process of integration took place after the completion of data analysis and entailed identifying similarities and differences, merging the results, and discussing the meaning of the integrated results across the two levels of analysis.

## 3. Results

### 3.1. Participants' Characteristics

Data for the quantitative part of the study was collected from a sample of 1368 HCWs (equivalent to a 99.9% response rate). The median age (IQR) was 33 (28, 43) years, and females comprised 60.1% of all respondents. Most of the respondents (76.5%) had either a certificate- or a diploma-level education and just above half (53.4%) had a working experience of fewer than six years. About three-quarters (77.5%) of the respondents were from government-owned health facilities, and Dar es Salaam constituted more than a quarter of all respondents (26.1%). About 42.1% of the respondents were working in district hospitals, and about two-thirds (62.6%) were working in the outpatient department. Nurses and midwives accounted for almost half (48.7%) of all respondents, followed by physicians (20.2%), Table 1.

**Table 1.** Participants' characteristics in the quantitative study (N = 1368).

| Variable | Frequency | Percentage |
|---|---|---|
| **Age (years)** | | |
| <30 | 470 | 34.4 |
| 30–39 | 483 | 35.3 |
| 40+ | 415 | 30.3 |
| **Sex** | | |
| Male | 546 | 39.9 |
| Female | 822 | 60.1 |
| **Education level** | | |
| Primary/secondary | 36 | 2.6 |
| Certificate | 437 | 31.9 |
| Diploma | 610 | 44.6 |
| Degree/master's | 285 | 20.8 |
| **Duration working** | | |
| <6 years | 731 | 53.4 |
| 6+ years | 637 | 46.6 |
| **Region** | | |
| Dar es salaam | 357 | 26.1 |
| Kilimanjaro | 187 | 13.7 |
| Lindi | 151 | 11.0 |
| Mbeya | 186 | 13.6 |
| Njombe | 158 | 11.5 |
| Simiyu | 137 | 10.0 |
| Tabora | 192 | 14.0 |
| **Health facility level** | | |
| Regional RH | 378 | 27.6 |
| District hospital | 576 | 42.1 |
| Health center | 414 | 30.3 |
| **Facility ownership** | | |
| Government | 1060 | 77.5 |
| Private/NGO/FBO | 308 | 22.5 |
| **Department** | | |
| Outpatient | 856 | 62.6 |
| Emergency | 248 | 18.1 |
| Inpatient | 45 | 3.3 |
| Pharmacy | 63 | 4.6 |
| Laboratory | 90 | 6.6 |
| Administration | 66 | 4.8 |
| **Cadre** | | |
| Physician | 277 | 20.2 |
| Nurse/midwife | 666 | 48.7 |
| Pharmacy | 96 | 7.0 |
| Laboratory | 137 | 10.0 |
| Admin/supporting staff | 99 | 7.2 |
| Others * | 93 | 6.8 |

* Other cadres included community health workers, counselors, data clerks/officers, medical attendants, nutritionists, physiotherapists, orthopedic technologists, and social welfare officers/workers.

Participants in the qualitative strand involved 100 HCWs, including 26 officials in the IDIs, and 74 were involved in FGDs/group interviews across four study sites (Dar es Salaam, Kilimanjaro, Tabora, and Simiyu regions). The participants' age ranged from 18 to 58, and they were all working at government health facilities (Table 2).

**Table 2.** Demographic characteristics of participants in the qualitative study.

| Variables | Frequency | Percentage |
|:---:|:---:|:---:|
| **Sex (*n* = 100)** | | |
| Male | 54 | 54.0 |
| Female | 46 | 46.0 |
| **Age in years (*n* = 99)** | | |
| 18–24 | 4 | 4.0 |
| 25–49 | 78 | 78.8 |
| 50+ | 17 | 17.2 |
| **Study site/region (*n* = 74)** | | |
| Dar es Salaam | 16 | 21.6 |
| Kilimanjaro | 17 | 23.0 |
| Tabora | 20 | 27.0 |
| Simiyu | 21 | 28.4 |
| **Education level (*n* = 99)** | | |
| Certificate | 20 | 20.2 |
| Diploma | 41 | 41.4 |
| Advanced diploma | 5 | 5.1 |
| Degree | 23 | 23.2 |
| Master's | 10 | 10.1 |
| **Cadre (*n* = 100)** | | |
| Nurses | 28 | 28.0 |
| Clinical officer | 7 | 7.0 |
| Doctor of medicine | 19 | 19.0 |
| Environmental health | 10 | 10.0 |
| Lab technicians | 13 | 13.0 |
| Pharmacists | 7 | 7.0 |
| Public health | 3 | 3.0 |
| Medical attendant | 8 | 8.0 |
| Sociologist | 2 | 2.0 |
| Dentist | 1 | 1.0 |
| Accountant | 1 | 1.0 |
| Statistician | 1 | 1.0 |
| **Marital Status (*n* = 100)** | | |
| Married | 72 | 72.0 |
| Separated | 4 | 4.0 |
| Widow | 2 | 2.0 |
| Single | 22 | 22.0 |
| **Position (*n* = 26)** | | |
| RMO | 3 | 11.5 |
| RVO | 3 | 11.5 |
| DMO | 7 | 26.9 |
| DVO | 8 | 30.8 |
| In charges | 5 | 19.2 |

RMO, Regional Medical Officer; RVO, Regional Vaccination Officer; DMO, District Medical Officer; DVO, District Vaccination Officer.

### 3.2. HCWs' COVID-19 Vaccine Uptake and Hesitancy by Selected Characteristics

The overall self-reported prevalence of COVID-19 vaccine uptake among HCWs in this study stood at 53.4%. The remaining were hesitant to vaccinate: 33.6% refused completely and 13.0% said that they would wait for some time to get vaccinated. The results further show statistically significant differences in vaccine uptake and hesitancy by HCWs' age, region, health facility level, facility ownership, duration working (years), perceived risk of COVID-19 infection, perceptions of COVID-19 vaccines safety and efficacy, and history of COVID-19 infection. The prevalence of vaccine uptake was highest among HCWs aged 40+ years (65.8%), in government-owned facilities (57.7%), and among physicians (60.3%). A long duration (6+ years) of service (64.7%) and a history of COVID-19 infection (60.4%) showed a higher prevalence of vaccine uptake (Table 3).

**Table 3.** HCWs' COVID-19 vaccine uptake and hesitancy by selected characteristics (N = 1368).

| Variables | Total | HCW Vaccinated | | | *p*-Value |
|---|---|---|---|---|---|
| | | Refused | Will Wait | Yes | |
| **Age (years)** | | | | | <0.001 |
| <30 | 470 | 199 (42.3) | 84 (17.9) | 187 (39.8) | |
| 30–39 | 483 | 158 (32.7) | 54 (11.2) | 271 (56.1) | |
| 40+ | 415 | 102 (24.6) | 40 (9.6) | 273 (65.8) | |
| **Sex** | | | | | 0.45 |
| Male | 546 | 190 (34.8) | 64 (11.7) | 292 (53.5) | |
| Female | 822 | 269 (32.7) | 114 (13.9) | 439 (53.4) | |
| **Education level** | | | | | 0.35 |
| Primary/secondary | 36 | 10 (27.8) | 6 (16.7) | 20 (55.6) | |
| Certificate | 437 | 162 (37.1) | 62 (14.2) | 213 (48.7) | |
| Diploma | 610 | 199 (32.6) | 74 (12.1) | 337 (55.2) | |
| Degree/master's | 285 | 88 (30.9) | 36 (12.6) | 161 (56.5) | |
| **Region** | | | | | <0.001 |
| Dar es salaam | 357 | 125 (35.0) | 55 (15.4) | 177 (49.6) | |
| Kilimanjaro | 187 | 57 (30.5) | 16 (8.6) | 114 (61.0) | |
| Lindi | 151 | 39 (25.8) | 19 (12.6) | 93 (61.6) | |
| Mbeya | 186 | 50 (26.9) | 49 (26.3) | 87 (46.8) | |
| Njombe | 158 | 66 (41.8) | 14 (8.9) | 78 (49.4) | |
| Simiyu | 137 | 47 (34.3) | 3 (2.2) | 87 (63.5) | |
| Tabora | 192 | 75 (39.1) | 22 (11.5) | 95 (49.5) | |
| **Health facility level** | | | | | 0.001 |
| Regional RH | 378 | 148 (39.2) | 56 (14.8) | 174 (46.0) | |
| District hospitals | 576 | 192 (33.3) | 80 (13.9) | 304 (52.8) | |
| Health centers | 414 | 119 (28.7) | 42 (10.1) | 253 (61.1) | |
| **Facility ownership** | | | | | <0.001 |
| Government | 1060 | 325 (30.7) | 123 (11.6) | 612 (57.7) | |
| Private/NGO/FBO | 308 | 134 (43.5) | 55 (17.9) | 119 (38.6) | |
| **Cadre** | | | | | 0.090 |
| Physician | 277 | 75 (27.1) | 35 (12.6) | 167 (60.3) | |
| Nurse/midwife | 666 | 232 (34.8) | 89 (13.4) | 345 (51.8) | |
| Pharmacy | 96 | 43 (44.8) | 9 (9.4) | 44 (45.8) | |
| Laboratory | 137 | 41 (29.9) | 18 (13.1) | 78 (56.9) | |
| Admin/supporting staff | 99 | 40 (40.4) | 12 (12.1) | 47 (47.5) | |
| Others | 93 | 28 (30.1) | 15 (16.1) | 50 (53.8) | |
| **Duration working** | | | | | <0.001 |
| <6 years | 731 | 300 (41.0) | 112 (15.3) | 319 (43.6) | |
| 6+ years | 637 | 159 (25.0) | 66 (10.4) | 412 (64.7) | 0.002 |
| **Ever been infected with COVID-19** | | | | | 0.020 |
| No | 960 | 340 (35.4) | 123 (12.8) | 497 (51.8) | |
| Yes | 328 | 88 (26.8) | 42 (12.8) | 198 (60.4) | |
| Don't know | 80 | 31 (38.8) | 13 (16.3) | 36 (45.0) | |
| **Total** | | **459 (33.6%)** | **178 (13.0%)** | **731 (53.4%)** | |

High hesitancy rates for vaccine uptake among HCWs was also supported by qualitative findings. For example, some of the HCWs leaders like the regional/council health management team (R/CHMT) members were among those who were hesitant to be vaccinated. However, they continued to distribute the vaccines and provide education to community members to promote acceptance and increase vaccine uptake.

The Intensive campaigns promoting vaccinations that were later implemented by the government and other stakeholders significantly contributed to observable changes in HCWs' vaccine acceptance and uptake.

IDI participants had the following to say about COVID-19 vaccine uptake and hesitancy:

*"HCWs had fears about COVID-19 vaccines. Those already vaccinated didn't accept the vaccine when offered initially. One day during a meeting, I joked with the District*

*Vaccination Officer (DVO), you DVO, you haven't been vaccinated up to now!, imagine, a COVID-19 vaccine coordinator! How can you coordinate?"(IDI R/CHMT, Dar es Salaam)*

*"It was until recently, that he agreed to get vaccinated and took a photo to show others that [he/she] has been vaccinated already. Generally, the uptake is not good, we still need to educate other staff [HCWs]." (IDI R/CHMT, Dar es Salaam)*

*"Generally, the uptake of COVID-19 vaccines among public servants is not good. Currently, some HCWs agree to get the Jab, however, the situation was not the same when the vaccines were introduced in the country. Also, non-health professionals such as teachers are still resisting vaccination." (IDI R/CHM, Dar es Salaam)*

*"The response, uptake, and understanding of COVID-19 vaccines was sub-optimal initially. In the first week, only four health workers were vaccinated, by the end of the month about 300 health workers in the district hospital were vaccinated. This happened after a lot of effort was put in place." (IDI HCW, Simiyu)*

*"Majority of the HCWs were not vaccinated until we came up with a campaign that aimed at raising awareness to the community about COVID-19 vaccines. We provided them with educational awareness about the positive side of vaccines. As a result, some of the HCWs got vaccinated. Initially, they did not take it positively, but after this education, 50% of them were vaccinated and even started to do outreach activities to promote vaccine uptake." (IDI R/CHMT, Dar es Salaam)*

During the FGDs, discussants pointed out that some of the HCWs accepted to take COVID-19 vaccines unwillingly because they were fearful of losing their jobs:

*"I decided to be vaccinated unwillingly. Otherwise, I would have waited at least three years to come. However, there was no way because I had already fallen in government hands." (FGD HCWs, Dar es Salaam)*

### 3.3. Factors Associated with COVID-19 Vaccine Uptake

Significantly higher odds of COVID-19 vaccine uptake was among HCWs aged 30–39 years (OR = 1.37, 95%CI 1.01, 1.86) and 40+ years (OR = 1.70, 95%CI 1.14, 2.55) than those <30 years; residing in Lindi (OR = 1.85, 95%CI 1.19, 2.87) and Tabora (OR = 1.66, 95%CI 1.11, 2.47) compared to Dar es Salaam region; working in district hospitals (OR = 1.68, 95%CI 1.23, 2.28) and health centers (OR = 1.92, 95%CI 1.41, 2.62) compared to regional/ referral hospitals; six or more years duration working at the health facility (OR = 1.73, 95%CI 1.25, 2.41); and perceived high/very high risk of COVID-19 infection (OR = 1.43, 95%CI 1.08, 1.88). Compared to physicians, all other cadres working in private/NGO/FBO-owned facilities had lower odds of vaccine uptake. Self-reported history of COVID-19 infection was not a significant predictor of vaccine uptake (OR = 1.23, 95% CI 0.93, 1.62, *p*-value = 0.14) in the adjusted analysis (Table 4).

Participants in IDIs and FGDs identified a variety of factors that promoted the uptake of COVID-19 vaccines. These included the following: education about safety, the importance of vaccines, and side effects; observing many deaths caused by COVID-19; campaigns for the house, i.e., households, churches, schools, and workplaces; and mass vaccination campaigns, where religious leaders and other influential people were vaccinated to motivate other people to believe that the vaccination is safe.

**Table 4.** Factors associated with COVID-19 vaccine uptake (N = 1368).

| Variables | COR (95%CI) | *p*-Value | AOR (95%CI) | *p*-Value |
|---|---|---|---|---|
| **Age (years)** | | | | |
| <30 | 1.00 | | 1.00 | |
| 30–39 | 1.93 (1.50, 2.50) | <0.001 | 1.37 (1.01, 1.86) | 0.05 |
| 40+ | 2.91 (2.21, 3.83) | <0.001 | 1.70 (1.14, 2.55) | 0.01 |
| **Sex** | | | | |
| Male | 1.00 | | | |
| Female | 1.00 (0.80, 1.24) | 0.98 | - | - |
| **Region** | | | | |
| Dar es salaam | 1.00 | | 1.00 | |
| Kilimanjaro | 1.59 (1.11, 2.28) | 0.01 | 1.38 (0.93, 2.04) | 0.11 |
| Lindi | 1.63 (1.11, 2.40) | 0.01 | 1.85 (1.19, 2.87) | 0.01 |
| Mbeya | 0.89 (0.63, 1.27) | 0.54 | 1.07 (0.72, 1.58) | 0.75 |
| Njombe | 0.99 (0.68, 1.44) | 0.96 | 1.09 (0.71, 1.67) | 0.69 |
| Simiyu | 1.77 (1.18, 2.65) | 0.01 | 1.32 (0.85, 2.06) | 0.22 |
| Tabora | 1.00 (0.70, 1.41) | 0.98 | 1.66 (1.11, 2.47) | 0.01 |
| **Health facility level** | | | | |
| Regional RH | 1.00 | | 1.00 | |
| District Hospitals | 1.31 (1.01, 1.70) | 0.04 | 1.68 (1.23, 2.28) | 0.001 |
| Health centers | 1.84 (1.39, 2.44) | <0.001 | 1.92 (1.41, 2.62) | <0.001 |
| **Facility ownership** | | | | |
| Government | 1.00 | | 1.00 | |
| Private/NGO/FBO | 0.46 (0.36, 0.60) | <0.001 | 0.37 (0.27, 0.52) | <0.001 |
| **Cadre** | | | | |
| Physicians | 1.00 | | 1.00 | |
| Nurse/midwife | 0.71 (0.53, 0.94) | 0.02 | 0.57 (0.41, 0.77) | <0.001 |
| Pharmacy | 0.56 (0.35, 0.89) | 0.01 | 0.54 (0.33, 0.89) | 0.02 |
| Laboratory | 0.87 (0.57, 1.32) | 0.51 | 0.85 (0.54, 1.33) | 0.47 |
| Admin/supporting staff | 0.60 (0.38, 0.95) | 0.03 | 0.59 (0.36, 0.98) | 0.04 |
| Others | 0.77 (0.48, 1.23) | 0.27 | 0.61 (0.37, 1.01) | 0.05 |
| **Duration working** | | | | |
| <6 years | 1.00 | | 1.00 | |
| 6+ years | 2.36 (1.90, 2.94) | <0.001 | 1.73 (1.25, 2.41) | 0.001 |
| **Perceived risk of COVID-19 infection** | | | | |
| No/low/medium risk | 1.00 | | 1.00 | |
| High/very high risk | 1.56 (1.22, 1.99) | <0.001 | 1.43 (1.08, 1.88) | 0.010 |
| **Ever been infected with COVID-19** | | | | |
| No | 1.00 | | 1.00 | |
| Yes | 1.45 (1.13, 1.86) | 0.004 | 1.23 (0.93, 1.62) | 0.140 |

COR, crude odds ratio; AOR, adjusted odds ratio, adjusted for age, region, health facility level, facility ownership, cadre, duration working at health facility (years), and perceived risk of COVID-19 infection.

> *"Education provided during different campaigns helped in promoting the uptake of COVID-19 vaccines among community members." (IDI R/CHMT Tabora MC)*

> *"Increased uptake of the COVID-19 vaccine was promoted by those already vaccinated to become good ambassadors in the community. Other factors that promoted uptake included the COVID-19 associated deaths, especially in wave 2 and wave 3, house-to-house campaigns, campaigns in churches, schools, and workplace." (Group interview, Kilimanjaro)*

*3.4. Barriers to COVID-19 Vaccine Uptake among HCWs in Tanzania*

Participants also reported on the barriers to COVID-19 vaccine uptake. The FGDs and IDIs generated the following barriers to COVID-19 uptake among HCWs. These included the following:

(1) Misinformation circulating in social and mainstream media about the safety and efficacy of COVID-19 vaccines and those spread by influential people such as religious leaders.

*"The factors which hindered [the] uptake of vaccines were social media and groups of people who tend to negatively discuss COVID-19 vaccines. They usually spread in groups through media/networks such as WhatsApp, Instagram, [and] Facebook. You would find strange things there and this caused problems to readers." (IDI R/CHMT Dar-es-Salaam)*

(2)      Lack of right information (inadequate knowledge) among HCWs on the different types of COVID-19 vaccines introduced in the country that have different dosing and timing.

*"We made a mistake in the way the COVID-19 vaccines were introduced. It was a very sudden and drastic move. HCWs were not prepared. They did not have the information or knowledge [they needed]. The COVID-19 vaccines came suddenly. For example, here they were received on Tuesday and launched on Wednesday. Even the vaccinators had no prior knowledge and at the very same time, we had already allowed other people to speak a lot on the vaccines. Things like that kill. The health care system had not provided enough education, when we come to [educate them], we realized that the community has already been misinformed. So, it was hard, and took a long time time for one to change because you then must change the mindset that was negative about the vaccines." (IDI R/CHMT Simiyu)*

(3)      The reported adverse events related to COVID-19 vaccines in other settings led to some worries among HCWs and were amplified by the statement in the initial COVID-19 vaccine consent form that said, "The government will not be responsible for any serious effects resulting from vaccine administration." In addition to the previous endorsement of the government on the use of local herbs to cure COVID-19, the statement caused fear to most Tanzanians as the government was perceived as if it was withdrawing itself from providing any support in case any problem happens.

*"I am not worried about JJ but the introduction of Sinopharm has made me feel worried because we got some news from the Coastal region [of Tanzania mainland] that there were serious adverse events reported that one of the people who got vaccinated got serious skin rashes and another one fainted." (IDI R/CHMT, Tabora)*

*"Majority of the HCWs, as well as community members, are so much worried about the consenting process especially the point which says that "the government will not be responsible for any serious effect caused by the vaccine." (IDI R/CHMT Tabora)*

(4)      The trust placed by political leaders and traditional healers in the use of traditional medicines for ameliorating COVID-19 symptoms also retarded the efforts in advocating for vaccine uptake.

*"There are people who still believe in local medication like 'steaming', and they have been convinced by traditional healers that it's the best medication, so these people are strongly opposing vaccines." (Group interview, HCW, Tabora)*

## 4. Discussion

The COVID-19 vaccine constitutes the ultimate intervention that is cost-effective in the struggle to eliminate COVID-19 infection. This study aimed at assessing COVID-19 vaccine uptake and hesitancy among HCWs in Tanzania. In the present study, the uptake of the COVID-19 vaccine was 53.4%, unlike existing evidence in low- and middle-income countries that show an uptake among HCWs to be as high as 82.5% [26]. The low uptake in our study could be a result of long-standing COVID-19 denial from the beginning of the pandemic to a few months before this study was conducted [27].

A significant difference in COVID-19 vaccine uptake by HCWs' age was observed in this study where individuals aged 40+ years carry a higher chance of uptake. This is not a new finding, as it was reported by other studies [2,6,9,10,28]. However, in our study, significant acceptance was observed from the age of thirty years, a much younger age than in other studies.

In this study, a significant association was found between the health facility level and the uptake of the COVID-19 vaccine with HCWs at the lower level having higher uptake than their counterparts. This suggests that the context under which HCWs perform their duties may influence their behaviors. The findings in the qualitative interviews allude to the influence of the context, such as continuous campaigns targeting both HCWs and community members. The influence of contextual factors in COVID-19 vaccine uptake has been reported in existing evidence, and some accounts addressing contextual factors may be among the quick wins to increase COVID-19 vaccine uptake [29].

In the present study, medical doctors had a significantly increased uptake of the COVID-19 vaccine compared to nurses. This finding is in keeping with the existing evidence that reported physicians to be more likely to take COVID-19 vaccination [29–32]. HCWs from public health facilities were more likely to report uptake of COVID-19 vaccines when compared to their counterparts in privately owned health facilities. This finding is in keeping with a study in Ghana [32]. The vaccination campaigns that have originated in public health facilities could have influenced HCWs' high uptake of COVID-19 vaccines. However, further studies could provide more light in this area to better understand the determinants of uptake and lessons that could be transferable in the two sectors.

HCWs with a perceived high/very high risk of COVID-19 infection were more likely to be vaccinated in the current study. Similar evidence has been documented elsewhere; increased susceptibility, perception, and fear of COVID-19 disease influenced vaccine acceptance in several other settings [33,34]. HCWs are generally at a higher risk of contracting COVID-19, as they are constantly exposed to infected patients. Linking this to the personal risk of acquiring the disease promoted the need for self-protection; hence, accepting vaccination becomes more plausible. Previous reviews reported perceiving the high risk of COVID-19 to have predicted compliance with preventive measures, including vaccination [35]. However, other studies report the role of survivorship bias from direct exposure to infected patients to be a hindrance against the HCWs' perceived need to be vaccinated [36].

The prevailing situation in the country concerning rampant misinformation regarding COVID-19 vaccination, as reported by qualitative findings, points out an important area that could change the status quo if deliberate interventions are devised and implemented. Hernandez and colleagues reported the proliferation of antivaccine social media that acts as a barrier to vaccine uptake among HCWs [34,37]. Similar to our findings, further evidence indicates that the infodemic is an important factor that hampers the COVID-19 uptake among HCWs and the general population [38]. Inadequate knowledge of the vaccine among HCWs was mentioned as one of the barriers to receiving the vaccine. Similarly, inadequate reliable information about the vaccine and its effectiveness was shown as a hindrance to vaccination in other settings [39]. According to Youssef et al., the recommendation by and confidence of relevant health authorities in and outside the country promotes people's confidence in the vaccine [39]. HCWs in our setting reported the initial passivity of the government toward COVID-19 vaccines and mixed information to have influenced vaccine hesitancy.

## 5. Methodological Considerations

Our study represents a comprehensive attempt to understand the determinants of COVID-19 vaccine uptake by using a mixed-methods design. Employing a multimethod approach in data collection, that is, a questionnaire, IDIs, group interviews, and FGDs provided a possibility of a comprehensive understanding of the study questions. However, the cross-sectional nature of the study limits causal relationships, and the results should be interpreted in that context. Although we tried to minimize errors by using trained research assistants (doctors and nurses), the possibility of social desirability and recall bias cannot be ruled out given the use of interviews as a method of data gathering.

## 6. Policy Recommendations

The findings in this study not only add to the existing evidence on the determinants of COVID-19 vaccine uptake among HCWs but also provide an avenue for recommendations for policy actions. First, a targeted campaign for HCWs should be developed and launched as a stand-alone, since the current approaches target the general population and do not look at the healthcare workforce as a vulnerable population to be targeted. These campaigns could also address the contextual and demographic variations (such as age, geographical zone, and level and ownership of health facility) in uptake observed in this study. Second, since HCWs are either members of professional bodies, associations, or trade unions such as the Medical Council of Tanganyika, Nursing and Midwives Council, Pharmacy Board, Medical Association of Tanzania, and Tanzania Union of Government and Health Employees, this could be used as a platform to mount intensive campaigns to promote the uptake of COVID-19 vaccine. Third, the Ministry of Health should look at different public health interventions that will increase understanding of COVID-19 vaccine knowledge among HCWs and the possibility of making the COVID-19 vaccine mandatory for HCWs who are in direct face-to-face contact with patients in their daily routines by developing and implementing guidelines for that.

## 7. Conclusions

This study represents the initial attempts in Tanzania set to understand the drivers of COVID-19 vaccine uptake and hesitancy among HCWs. The findings suggest that a considerable number of HCWs are still unvaccinated against COVID-19 infections. The significant determinants of vaccine uptake are the age of 40 years and above, region of residence, health facility level, and ownership of the health facility where one is working. A difference in uptake between physicians and nurses was observed, and perceptions of safety and efficacy influenced uptake. The predominance of contextual influence on COVID-19 vaccine uptake observed in this study calls for interventions to focus on addressing contextual determinants. Moreover, the low uptake among young people (HCWs) who are the majority demographically in Tanzania and who equally spread COVID-19 to others in the community requires further exploration for a tailored intervention. We recommend further studies for assessing interventions that consider culture and ethnicity.

**Author Contributions:** Conceptualization, M.A.A., H.P.N., B.J.N., N.K., A.K., E.M., L.E.M., I.B.M. and S.E.M.; Methodology, M.A.A., A.K., I.B.M., L.E.M., K.K.N., S.E.M., A.K., E.M., B.S. and E.H.S.; Investigation, M.A.A., I.B.M., L.E.M., F.N., K.K.N., H.P.N., B.J.N., S.E.M., A.K., E.M., E.H.S., B.S. and J.T.K.; Analysis, I.B.M., M.A.A., L.E.M., K.K.N. and A.K.; Writing Original Draft, M.A.A., H.P.N., I.B.M., J.T.K. and A.K.; Review and Editing, M.A.A., I.B.M., H.P.N., J.T.K., E.M., B.J.N., K.K.N., L.E.M., E.H.S., F.N., N.K., B.S., S.E.M. and A.K.; Funding Acquisition, M.A.A., N.K., F.N. and S.E.M.; Resources: M.A.A., S.E.M. and F.N.; Supervision, M.A.A., J.T.K., S.E.M., B.S., E.H.S., E.M., I.B.M., K.K.N. and A.K. All authors have read and agreed to the published version of the manuscript.

**Funding:** This work was supported by the United Nations Children's Fund (UNICEF) Tanzania (TZA/PCA2020104/HPD2020158).

**Institutional Review Board Statement:** Ethical approval to conduct this study was obtained from the Muhimbili University of Health and Allied Sciences Ethical Review Board, Institutional Review Board (IRB) reference number MUHAS-REC-08-2021-839. Permission to collect data in Regions and Councils was obtained from the President's Office, Regional Administration, and Local Government (PORALG), the Ministry of Health (MoH), Regional Secretariat (RS), and Local Government Authorities (LGAs). Unique identification numbers were used instead of names to safeguard participant information. All interviews were conducted in a separate and quiet room within each health facility for the privacy and confidentiality of participant information.

**Informed Consent Statement:** Informed consent was obtained from all subjects involved in the study.

**Data Availability Statement:** The dataset that was used in this study is available upon request to the corresponding author.

**Acknowledgments:** The authors acknowledge their corresponding institutions for providing support to conduct the study. We recognize the cooperation of all the HCWs who took part in this study. We acknowledge the support from the regional and district medical and vaccine officers and the medical officers in charge for permission to conduct this study despite the challenges caused by the COVID-19 pandemic. We thank our research assistants for their dedication to conducting this study promptly: Melina Mgongo, Doris Mbata, Oko Okong'o, Zenaice Aloyce, Martha Joseph, Mtumwa Bakari, Nyanjura Manyama, Zenais Kiwale, Anastazia Ngowi, Barikiel Panga, Novatus Tesha, Albert Majura, Naike Nathaniel, Julietha Tibyesiga, Loveness Kimaro, Judith Kokuleba, Ngusa Kalambo, Constancia F Luyenga, Monica Mtei, Jackline Ngowi, Edson B Jeremiah, Witness Simon, Erick Kazoka, and Chrispin Mgute.

**Conflicts of Interest:** The authors declare no conflict of interest.

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
