# Peer review of "Determinants of COVID-19 Vaccine Uptake and Hesitancy among Healthcare Workers in Tanzania: A Mixed-Methods Study"

_covid, doi:10.3390/covid3050058_

Round 1
Reviewer 1 Report
1. The abstract Methods: Please provide sampling techniques, research year, main used questionnaires, and statistical software concisely and clearly.
2. Please provide information about validation process of questionnaire
3. Amend p-value of <0.05 : p-value of ≤0.05
4. How integration of qualitative and quantitative was conducted
5. Compare the quantitative results with qualitative results
Minor editing of English language required
Author Response
Please see the attached report

Reviewer 2 Report
1. Introduction focuses on Tanzania which is also stated in the abstract.
2. This was a mixed methods design and the components defined.
3. Method of sampling and size presented.
4. Study instruments need to be defined and scoring shared. For example, what are certificate and diploma levels of education? What is a clinician? Is that another word for physician? Because this is an international journal, all readers need to be able to understand your variables and how measured. Be careful that variables are consistently defined. Age 40 years is not old age as is stated in conclusions. The instrument section is greatly lacking.
5. In the discussion, please share how your data compare to the literature.
6. What are your study’s limitations? Did you meet your sample size?
7. How does this study add to the literature? This is critical because so many papers have been written about the pandemic. What makes your work original? Why is it needed?
1.
The English is ok. The word data is plural and thus needs a plural verb. The variables need to be defined.
Author Response
Please see the attached report

Reviewer 3 Report
we read with interest the article by Amour et al discussing Determinants of COVID-19 vaccine uptake and hesitancy among health care workers in Tanzania: a mixed methods study which is an interesting topic impacting third world countries. The work is highly original and has high scientific merit. It is quite well constructed and summarizes in a clear form the results of the study
there are a few minor comments that would suggest to the authors:
1- There is need to include a statistical power analysis for the study and the number of participants included
2- “Was there any documented hesitancy related to different types or brands of COVID vaccine? (messenger RNA (mRNA) vaccines, viral vector vaccines, protein subunit vaccines, whole virus vaccines)
3-Conclusion and Discussion: Please expand to include a statement about the need for further research regarding ethnicity and culturally sensitive interventions.”
4-“The work should have a future perspective of the implication of the outcome of this work and can public health intervention be applied to affect this vaccination acceptance”
Author Response
Please see the attached report.
